# Unified Multi-Dataset OCT Learning with RETFound–CNN–FNN

**Jarin Tasnim**                                        2005083@CSE.BUET.AC.BD

**Abid Hasan Khondaker**                                2005063@CSE.BUET.AC.BD

*Department of CSE, Bangladesh University of Engineering and Technology (BUET)*

## Abstract

Early diagnosis of retinal disease is important for preventing permanent vision loss. Many deep learning systems are trained on only one dataset, which often reduces their ability to generalize in practice. We present a unified multi-task learning framework trained on three heterogeneous OCT datasets (Duke, Zhang, and OLIVES). RETFound is used as a shared backbone, followed by a CNN neck that restores local spatial information, and then dataset-specific FNN heads. We train with source-aware loss masking and progressive unfreezing to improve stability. Our model achieves very high disease classification accuracy (Duke 99.59%, Zhang 99.38%, OLIVES 99.9%) and strong multi-label biomarker detection performance.

## 1. Introduction & Related Work

Optical Coherence Tomography (OCT) is widely used to diagnose retinal diseases such as AMD, DME, and CNV. CNNs are strong at capturing local image patterns, while Vision Transformers (ViTs) are good at modeling global context. However, models trained on a single dataset often do not transfer well to other datasets or label settings. Multi-task learning can improve robustness, but it is still difficult to train one model across datasets with different label definitions.

To address this, we combine RETFound (a retinal foundation model) with a CNN-to-FNN feature pipeline. Our main contributions are: (1) one unified MTL framework for Duke, Zhang, and OLIVES; (2) a CNN neck that brings back spatial inductive bias after ViT tokenization; and (3) a dataset-aware loss-masking strategy that reduces cross-task interference during training.

## 2. Datasets & Preprocessing

We use three datasets so the model learns both disease-level labels and fine-grained biomarker labels:

- **Duke** (1): 3,231 samples (AMD, DME, Normal).

- **Zhang (Kermany)** (2): 84,000 samples (CNV, DME, Drusen, Normal).

- **OLIVES** (4): 78,822 samples (16 biomarkers, 2 disease labels).

All images are resized to $224 \times 224$, converted to 3 channels for ViT compatibility, normalized, and split into 70/15/15 train/validation/test sets using stratified random sampling to ensure balanced class distributions and prevent data leakage. Zhang retains its original predefined splits. We apply mild data augmentation during training (random rotations, brightness adjustments).

## 3. Methodology

Our pipeline has three stages: a shared backbone, a CNN neck, and task-specific heads:

$$\text{Input} \rightarrow \text{RETFound ViT-L/16} \rightarrow \text{CNN Neck} \rightarrow \text{FNN Heads}$$

**Backbone & CNN Neck:** We leverage RETFound, a powerful Vision Transformer (ViT-L/16) pre-trained on large-scale retinal imaging datasets, as our shared backbone. RETFound outputs tokens $X \in \mathbb{R}^{1 \times 197 \times 1024}$. We reshape the patch tokens into a spatial grid $\mathbb{R}^{14 \times 14 \times 1024}$ to recover local structure that is useful for small retinal findings. A two-layer CNN ($3 \times 3$ Conv followed by $1 \times 1$ Conv) refines these features.

**Dataset-Specific FNN Heads:** The shared features are passed to separate output heads: Duke (3-class softmax), Zhang (4-class softmax), and OLIVES (16-label sigmoid for biomarkers plus 2-class softmax for diseases).

**Training Strategy:** We use Cross-Entropy for disease classification and scaled Binary Cross-Entropy for biomarker prediction. Loss masking updates only the head relevant to the current dataset batch, which reduces gradient conflicts. We also use progressive unfreezing (keeping blocks 0–18 frozen at the start, then gradually unfreezing) to protect useful pre-trained representations.

## 4. Experiments & Results

**Experimental Setup:** We conducted five independent training runs with stratified validation. High accuracies are expected because the retinal classes are visually distinct, RETFound is a strong retinal backbone, and the 70/15/15 splits prevent train/test contamination. RETFound without fine-tuning achieved 44.4% accuracy on Duke, confirming that the final gains come from dataset-specific adaptation rather than raw backbone transfer.

**Ablation Study:** Table 1 shows that the full pipeline improves over RETFound alone. The CNN neck, loss masking, and progressive unfreezing together yield the strongest Duke gain (+1.65%). Isolating the exact requirement of progressive unfreezing itself is future work.

Table 1: Ablation study: component contributions.

| Configuration | Duke (Acc) | Zhang (Acc) | OLIVES Disease (Acc) | Avg Improvement |
|---|---|---|---|---|
| RETFound baseline | 97.94% | 99.38% | 99.84% | — |
| Full method (+ CNN neck, loss masking, progressive unfreezing) | 99.59% | 99.38% | 99.90% | +0.55% |

Table 2 summarizes disease classification performance across all datasets.

Table 2: Disease classification summary on test data.

| Task | Accuracy | Macro-F1 | Weighted-F1 |
|---|---|---|---|
| Duke (3-class) | 99.59% | 0.9951 | 0.9959 |
| Zhang (4-class) | 99.38% | 0.9938 | 0.9938 |
| OLIVES disease (2-class) | 99.90% | 0.9999 | 0.9999 |

**Duke & Zhang:** The confusion matrices in Figures 1 and 2 show clear class separation; Zhang errors are mostly between DRUSEN and CNV, which is clinically reasonable.

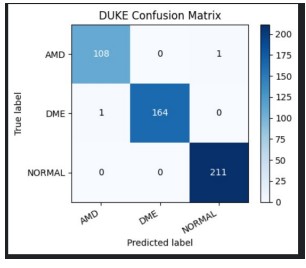

(a) Duke Matrix

**[DUKE] Classification report**

|  | precision | recall | f1-score | support |
|---|---|---|---|---|
| AMD | 0.9908 | 0.9908 | 0.9908 | 109 |
| DME | 1.0000 | 0.9939 | 0.9970 | 165 |
| NORMAL | 0.9953 | 1.0000 | 0.9976 | 211 |
| accuracy |  |  | 0.9959 | 485 |
| macro avg | 0.9954 | 0.9949 | 0.9951 | 485 |
| weighted avg | 0.9959 | 0.9959 | 0.9959 | 485 |

(b) Duke Report

Figure 1: Duke Classification Performance.

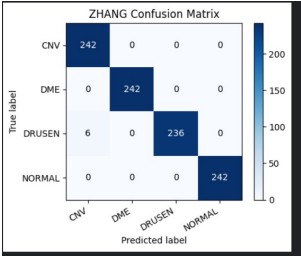

(a) Zhang Matrix

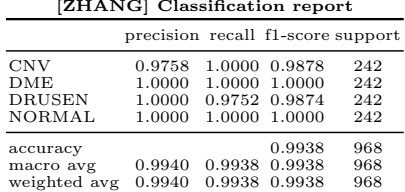

**[ZHANG] Classification report**

|  | precision | recall | f1-score | support |
|---|---|---|---|---|
| CNV | 0.9758 | 1.0000 | 0.9878 | 242 |
| DME | 1.0000 | 1.0000 | 1.0000 | 242 |
| DRUSEN | 1.0000 | 0.9752 | 0.9874 | 242 |
| NORMAL | 1.0000 | 1.0000 | 1.0000 | 242 |
| accuracy |  |  | 0.9938 | 968 |
| macro avg | 0.9940 | 0.9938 | 0.9938 | 968 |
| weighted avg | 0.9940 | 0.9938 | 0.9938 | 968 |

(b) Zhang Report

Figure 2: Zhang Classification Performance.

**OLIVES Biomarkers:** Disease classification is very high, but multi-label biomarker prediction remains harder. Micro-F1 is 0.8209 while Macro-F1 is 0.3997, reflecting strong class imbalance, especially for rare biomarkers.

## 5. Conclusion

We presented a hybrid Transformer-CNN multi-task model for OCT analysis across Duke, Zhang, and OLIVES. The shared RETFound backbone, CNN neck, and dataset-specific heads provide strong disease classification results on all three datasets. Biomarker prediction remains more challenging under heavy class imbalance, and cross-dataset transfer experiments plus a dedicated study of progressive unfreezing are left for future work.

**Code Availability.** The notebooks used for training and testing are available at https://github.com/abid5063/ML-472-project.

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
