# OpenReview forum: "Unified Multi-Dataset OCT Learning with RETFound-CNN-FNN"
_MIDL.io/2026/Short_Papers — MIDL 2026 - Short Papers Poster_

### Official Review · Reviewer_Wy1M · 2026-05-07
**OCT learning**

**Rating:** 3
**Confidence:** 4

**Review:**

The paper presents a useful engineering contribution with promising empirical performance and practical relevance for unified retinal OCT analysis. However, the current version would benefit significantly from stronger experimental validation, deeper ablation analysis, more rigorous handling of class imbalance, and clearer evidence that the method generalizes beyond standard train/test splits.

**Summary:**

This paper presents a unified multi-task learning framework for retinal OCT analysis across three heterogeneous datasets: Duke, Zhang , and OLIVES. The proposed architecture combines the retinal foundation model RETFound with a CNN neck and dataset-specific FNN heads, enabling the system to leverage both global contextual representations from Vision Transformers and local spatial features important for retinal pathology detection.

**Strengths:**

Addresses a clinical problem: robust retinal OCT analysis across heterogeneous datasets.
Proposes a unified multi-task framework capable of handling multiple datasets with different label spaces.
Effectively combines Transformer-based global feature extraction (RETFound) with CNN-based local spatial refinement.
Includes both disease classification and biomarker detection tasks, increasing the scope and applicability of the work.

**Weaknesses:**

Limited methodological novelty since the framework mainly combines existing components and training strategies.
Missing ablation studies to isolate the contribution of the CNN neck, loss masking, and progressive unfreezing.
Lacks comparison with strong recent baselines and prior multi-task OCT methods.
Extremely high reported accuracies raise concerns about possible dataset leakage or overly easy train/test splits.
No external validation or cross-dataset generalization experiments are provided.

**Justification Of Rating:**

I think this paper has some interesting engineering contribution with promising practical relevance, but the experimental validation and novelty are currently insufficient for a stronger recommendation.

---

### Decision · Program_Chairs · 2026-05-08

Accept (Poster)